# Speckle Tracking Echocardiography in Patients with Non-Ischemic Dilated Cardiomyopathy Who Undergo Cardiac Resynchronization Therapy: A Narrative Review

**DOI:** 10.3390/diagnostics14111178

**Published:** 2024-06-03

**Authors:** Nikolaos Antoniou, Maria Kalaitzoglou, Lamprini Tsigkriki, Amalia Baroutidou, Adam Tsaousidis, George Koulaouzidis, George Giannakoulas, Dafni Charisopoulou

**Affiliations:** 1Cardiology Department, General Hospital G. Papanikolaou, 57010 Thessaloniki, Greece; nikosn771@gmail.com (N.A.); mariakalaitzo25@gmail.com (M.K.); linatsigr@gmail.com (L.T.); adamtsaou@hotmail.com (A.T.); 2Cardiology Department, AHEPA University General Hospital, 54636 Thessaloniki, Greece; bamalia27@gmail.com (A.B.); g.giannakoulas@gmail.com (G.G.); 3Department of Biochemical Sciences, Pomeranian Medical University, 70-204 Szczecin, Poland; 4Great Ormond Street Hospital, London WC1N 3JH, UK

**Keywords:** non-ischemic dilated cardiomyopathy, cardiac resynchronization therapy, speckle tracking echocardiography, strain, strain rate, dyssynchrony

## Abstract

Non-ischemic dilated cardiomyopathy (DCM) represents a significant cause of heart failure, defined as the presence of left ventricular (LV) dilatation and systolic dysfunction unexplained solely by abnormal loading conditions or coronary artery disease. Cardiac resynchronization therapy (CRT) has emerged as a cornerstone in the management of heart failure, particularly in patients with DCM. However, identifying patients who will benefit the most from CRT remains challenging. Speckle tracking echocardiography (STE) has garnered attention as a non-invasive imaging modality that allows for the quantitative assessment of myocardial mechanics, offering insights into LV function beyond traditional echocardiographic parameters. This comprehensive review explores the role of STE in guiding patient selection and optimizing outcomes in CRT for DCM. By assessing parameters such as LV strain, strain rate, and dyssynchrony, STE enables a more precise evaluation of myocardial function and mechanical dyssynchrony, aiding in the identification of patients who are most likely to benefit from CRT. Furthermore, STE provides valuable prognostic information and facilitates post-CRT optimization by guiding lead placement and assessing response to therapy. Through an integration of STE with CRT, clinicians can enhance patient selection, improve procedural success rates, and ultimately, optimize clinical outcomes in patients with DCM. This review underscores the pivotal role of STE in advancing personalized management strategies for DCM patients undergoing CRT.

## 1. Introduction

Dilated cardiomyopathy is defined by the presence of LV dilatation and systolic dysfunction unexplained solely by abnormal loading conditions or CAD [1,2]. DCM is prevalent in 36 patients per 100,000 population, accounting for 40% of all cardiomyopathies [3,4]. According to recent data from a large observational study in northern Europe, DCM accounts for heart failure (HF) in 7.9% of unselected patients admitted to the hospital for HF [4]. Heart failure is the leading cause of illness and death in individual DCM.

In DCM, the most prevalent phenotypes of HF are either HF with a mid-range ejection fraction or HF with a reduced ejection fraction [5,6,7]. As the disease progresses, medication options become limited, and heart failure worsens. This severely limits the patient’s ability to perform even the smallest of daily tasks. However, patients suffering from DCM, be it of ischemic or non-ischemic etiology, have been found to benefit significantly from cardiac resynchronization therapy (CRT) [8,9]. CRT has proven superior to conventional medical treatment by decreasing mortality and morbidity [8,9].

Over the years, multiple echocardiographic parameters have been utilized to evaluate mechanical dyssynchrony of the LV, defined as the difference in the timing of mechanical contraction between different segments of the LV [10]. The assessment of LV mechanical dyssynchrony can be performed using many approaches, including tissue Doppler imaging, speckle tracking echocardiography, or cardiac magnetic resonance imaging (cMRI). Parameters include measures of interventricular or intraventricular mechanical delay, such as the temporal variance between the peak strain or strain rate in different segments of the left ventricle [10,11]. The PROSPECT (Predictors of Response to Cardiac Resynchronization Therapy) trial evaluated several echocardiographic indices, using both conventional and tissue Doppler imaging methods, such as interventricular and intraventricular delay and the temporal variance in the peak strain and strain rate in different segments of the LV, in order to predict a favorable response to CRT [12]. However, the trial found that no single echocardiographic measurement of dyssynchrony can be recommended for improving patient selection for CRT. This is because many clinical, electrocardiographic, and echocardiographic features can affect the CRT response.

Large randomized clinical trials have shown that CRT can improve symptoms, reduce hospital admissions, and prolong survival in some patients [13,14,15]. However, up to 30% of carefully selected patients do not benefit from this costly and invasive intervention [16,17]. One proposed benefit mechanism from CRT is correcting mechanical dyssynchrony within the LV [16,17]. However, the PROSPECT trial found that no single echocardiographic measurement of dyssynchrony can be recommended for improving patient selection for CRT [12]. This is because many clinical, electrocardiographic, and echocardiographic features can affect the CRT response.

Newer imaging methods such as speckle tracking echocardiography (STE) in two-dimensional (2D) enable clinicians to detect changes in myocardial function, measured in terms of strain and strain rate [18,19]. Myocardial strain, or deformation, is the percentage change in length of a myocardial segment relative to its length at the baseline; the strain rate is the rate of deformation (strain per unit of time). This technique offers an angle-independent, easy, and fast measurement of the strain. Compared with the tissue Doppler strain, STE has a better correlation with cardiac magnetic resonance imaging (cMRI), improved feasibility, and less interobserver and intraobserver variability [20]. Although this ultrasonographic modality appears to have a variability in methods and normal values from hardware and software difference and can be affected by the afterload, it still remains vital in assessing the severity of HF in patients with DCM and its progression [21]. Therefore, the assessment of LV mechanics through STE in 2D in these patients constitutes a useful tool in terms of CRT.

The application of STE in three-dimensional (3D) represents a notable progress in evaluating myocardial mechanics. This approach enables a more thorough assessment of heart function by gathering volumetric data and offering insights into myocardial deformation in all spatial dimensions. Research has shown that 3D-STE is more effective than standard 2D approaches in precisely analyzing the myocardial strain and detecting minor anomalies in heart mechanics [22].

Comparative examinations of data generated from 3D and 2D approaches have revealed the benefits of 3D STE. The comparisons demonstrate enhanced precision and consistency in strain assessments using 3D imaging, especially in intricate heart shapes and areas with substantial myocardial deformation [23]. In addition, 3D STE improved the imaging and measurement of the myocardial strain across the whole cardiac cycle, providing superior diagnostic capabilities and a prognostic value in different cardiac diseases.

This is a narrative review aimed at evaluating the clinical value of STE in patients with non-ischemic DCM who undergo cardiac resynchronization therapy.

## 2. Speckle Tracking in Non-Iscemic Dilated Cardiomyopathy

### 2.1. STE and Prediction of CRT Response

In a study conducted at a time when stain imaging was a new technique, Carasso et al. investigated whether specific LV strain patterns identified through echocardiography can serve as prognostic indicators for CRT outcomes, independent of conventional timing parameters [24]. The study included 76 consecutive patients who underwent CRT device implantation and whose echocardiograms were analyzable (Table 1). The time interval between baseline and postprocedural echocardiographic studies was 6 ± 4 months. At the end of the study, 44 patients were identified as responders, while 32 were classified as non-responders. The peak average longitudinal strain showed abnormally low values in all segments for both responders and non-responders, indicating their underlying left ventricular dysfunction. However, the non-responder group’s baseline average segmental peak strain was notably lower. The septal wall strain exhibited similarity in both groups, while strains in the lateral, inferior, and anterior regions were higher at the baseline among responders. Further analysis was conducted based on pre-and post-procedural strain curve patterns. In 88% of non-responders, at least one segment exhibited passive characteristics, defined as a continuous severe shortening (peak strain ≥ +5%) throughout the systole. This pattern predominantly occurred at the basal lateral segment but could also manifest in any other segment. Conversely, only 2% of responders had a single passive segment. The absence of passive segments was found to be 98% sensitive and 88% specific for the response, with positive and negative predictive values of 91% and 97%, respectively. This study highlights that LV strain patterns, as assessed by echocardiography, hold a predictive value in determining the likelihood of a positive response to CRT among DCM patients. Importantly, the study suggests that beyond the timing of mechanical activation, LV strain patterns offer additional predictive insights into CRT efficacy. The main limitation of the study is the small sample size, which may limit the generalizability as well as the complexity and feasibility of the echocardiographic techniques.

In another early study, Phillips et al. examined the influence of longitudinal rotation (LR) on myocardial systolic velocity, Table 1 [25]. Longitudinal rotation was assessed in the apical 4-chamber view by speckle-tracking techniques, and myocardial systolic velocities for basal septum and lateral LV were analyzed from tissue Doppler images. The quartiles of LR distribution were analyzed for differences in their systolic velocities. A major finding of this study was that longitudinal cardiac rotation alters the profile and amplitude of basal septal and lateral systolic myocardial velocities. However, velocity time-based measures were not predictive of CRT response. The findings are limited due to a lack of long-term follow-up data.

In a study conducted by Mele et al., the effectiveness of different echocardiographic parameters was explored to identify patients who respond positively to CRT, Table 1 [26]. The standard deviation of the averaged time-to-peak longitudinal negative strain (Tε-SD) and positive systolic velocity (Tv-SD) of 12 LV segments were calculated before and after 6 months of CRT. The researchers performed a comparative analysis to determine which method is more reliable in predicting CRT responders. On univariable analysis, baseline Tε-SD and Tv-SD were both significantly associated with CRT response. However, the area under the receiver operating characteristic curve was better for Tε-SD. Only Tε-SD retained an independent prognostic value for CRT response on a bivariable analysis. The study’s findings indicate that myocardial deformation and velocity dyssynchrony parameters are useful in identifying responders to CRT. Nevertheless, the researchers observed that myocardial deformation analysis might offer certain advantages over velocity dyssynchrony assessment in predicting CRT response. The logistic models were adjusted for etiology, but the analysis results did not change. Potential bias due to difficult echocardiographic techniques is present.

Sipula et al. conducted a study to evaluate the relationship between speckle tracking and the CRT response and the possibility of using strain values to optimize CRT performance in patients who did not respond well to the initial implantation [27]. The study assessed four types of strains—longitudinal, circumferential, radial, and area—in 31 patients with IDCM and 29 patients with DCM (Table 1). Out of the total number of patients, 21 were non-responders, and 39 were responders after the first intervention. The echocardiographic parameters revealed that LVEF was a strong predictor of NYHA improvement. However, area strain (a combination of longitudinal and circumferential strains) was found to be the best predictor of NYHA improvement among the strain parameters. After 3 months, optimization was performed in 19 patients, excluding 1 who refused and 1 who was lost from the follow-up. The researchers optimized the atrioventricular interval concerning LV filling and the interventricular interval according to all heart strains for all non-responders. Out of the non-responders who underwent optimization, 7 responded well, while 12 remained non-responsive. Out of the 19 optimized patients, 9 had DCM, and 10 had CAD. Following the optimization, 8 out of 9 DCM patients responded positively, while only 3 out of 10 CAD patients showed an improvement. Despite the small sample size, this effect was statistically significant (*p* = 0.02). The study’s potential limitations may stem from the small sample size, limiting the generalizability of its findings, while the complexity of the measurements potentially introduces bias into the results.

In a study conducted by Cimino et al., they examined the clinical and echocardiographic characteristics that are related to a positive response to CRT in patients with DCM [28]. According to current guidelines, the study included 24 patients who were recommended for CRT implantation (Table 1). The patients underwent echocardiography screening before the implantation and 6 months after. The echocardiography analysis included 2D-speckle tracking, real-time three-dimensional echocardiography, and color TDI. The cardiac dyssynchrony was assessed using Doppler, color TDI, and 3D echocardiography. CRT response was defined as a decline in LVESV > 10% after 6 months from the CRT implantation. After the 6-month follow-up, 12 patients (50%) responded favorably to CRT (CRTR+). In the CRTR+ patients, the LVEF and GLS improved significantly. Specifically, the LVEF increased from 29 ± 4 at baseline to 35 ± 10 after CRT implantation (*p* = 0.002). Also, in 6 months, the GLS was −12 ± 4, while at the baseline, the GLS was −10 ± 2 (*p* = 0.003). Furthermore, the GLS before implantation in CRTR+ patients was higher than in CRTR- patients (−10 ± 2% vs. −8 ± 2%; *p* = 0.03). The authors concluded that patients who did not respond favorably to CRT had greater LV volumes, lower LVEF, more reduced GLS, greater RV impairment, and higher pulmonary artery pressure at the baseline. However, in the multivariable analysis, the LVEF calculated from 3D volumes was the stronger predictor of CRT response. A cutoff value of the LVEF of 22.15% significantly predicted the response to CRT with a sensitivity of 80% and a specificity of 50%.

On the other hand, Pescariu et al. investigated if several echocardiographic parameters, especially strain imaging, could be related to CRT functional parameters such as sensing and pacing [29]. The study group consisted of 48 patients with DCM, while 64.6% of them had been diagnosed with DCM (Table 1). They documented significant correlations between the global longitudinal strain (GLS), posterolateral strain (PLS), EF and LV sensing (respectively, *p* < 0.001) and LV pacing parameters (*p* < 0.001). Pescariu and colleagues showed that increased LV-GLS and LAV values are linked with an increased LV sensing threshold, whereas decreased LV-GLS, RV-GLS, and TAPSE increased the pacing threshold. An important finding is that focal GLS, specifically LV-PLS, could help predict a positive CRT response because it assesses the strain in the region where the coronary sinus lead is positioned. According to the results of this study, global and focal strain imaging techniques could help guide lead positioning and device programming, increasing the chances of a positive CRT response. The main limitation is the small patient population. The authors did not report precisely when the follow-up echocardiography studies after the CRT implantation were performed.

### 2.2. STE and Diastolic Dyssynchrony after CRT

In a recent study, Gurgu et al. aimed to identify specific parameters of diastolic dyssynchrony that could help predict the response to CRT [30]. The study included 62 patients diagnosed with DCM and recommended for CRT-P (Table 1). Of these patients, 29% had type III diastolic dysfunction, 63% type II, and 8% type I. The study evaluated diastolic dyssynchrony before and after CRT implantation using offline speckle-tracking-derived TDI timing assessment of the simultaneity of E″ and A″ basal septal and lateral walls in a 4-chamber view. The authors introduced two new parameters (E″T and A″T) that measured the time difference between the septal and lateral wall E″ and A″ peaks, respectively, before and after resynchronization therapy. The patients were categorized into three groups: responders, super-responders, and non-responders. Only three patients (5%) were non-responders. In these three patients, greater baseline E″T and A″T values were recorded. The study revealed that E″T > 80 ms (AUC = 0.8989, 95% CI 0.8401 to 0.9577, *p* < 0.0001) and A″T > 30 ms (AUC = 0.8938, 95% CI 0.8351 to 0.9526, *p* < 0.0001) are the cut-off values associated with a CRT response. The follow-up period was 45 ± 19 months. The study’s main limitation is the small number of patients, especially the low number of non-responders. However, the small number of non-responders is due to a highly selective population included in the study and a close and tailored follow-up for each patient to maximize the CRT response. The study’s limitations include the small sample size, but also the complexity in measuring diastolic dyssynchrony, which may introduce bias or error into the results.

### 2.3. STE and Right Ventricle

D’Andrea et al. conducted a study to investigate the effects of CRT on the function of the right ventricle (RV) in patients with DCM who do not show signs of right heart failure [31]. The study used 2D strain analysis of the RV longitudinal strain in RV septal and lateral walls to evaluate the impact of CRT on the RV myocardial strain over 6 months. The research involved 110 participants with either idiopathic (*n* = 60) or ischemic (*n* = 50) DCM (Table 1). The study observed that RV diameters were slightly increased in patients with idiopathic DCM, while RV tricuspid annulus systolic excursion and Tei-index were similar in both groups. The RV global longitudinal strain and regional peak myocardial strain were significantly impaired in patients with idiopathic DCM compared to those with ischemic DCM. The study identified 70 patients (63.3%) who were long-term responders to CRT using a left ventricular end-systolic volume as a marker for the response. Ischemic DCM patient responders to CRT showed a significant improvement in the RV peak systolic strain. On the other hand, no improvement in the RV function was observed in patients with idiopathic DCM and in ischemic patients who did not respond to CRT. Multivariable analysis revealed that in the overall population, ischemic etiology of DCM, a positive response to CRT, and longitudinal intraventricular dyssynchrony were the only independent determinants of RV global longitudinal strain after CRT. The study also found that in idiopathic DCM, RV deformation remained impaired even after the improvement in loading conditions during CRT, indicating the presence of a more significant involvement of the RV structure and function in the myopathic process in patients with idiopathic DCM. Similar to other studies, there is a limited generalizability of the results due to the observational nature of the study.

Vitarelli et al. investigated whether different LV and RV parameters of STE could predict the response to CRT [32]. A total of 81 patients were recruited; 49 patients with dilated cardiomyopathy of the ischemic etiology and 32 with DCM [19]. After 6 months, 50 (62%) patients responded favorably to CRT. Through their investigation, the authors find that incorporating RV dyssynchrony indexes enhances the predictive capability of CRT success beyond LV dyssynchrony metrics alone. This suggests that assessing both LV and RV dyssynchrony could provide a more comprehensive evaluation of patient suitability for CRT. The study underscores the potential significance of RV dyssynchrony evaluation in optimizing patient selection for CRT, offering valuable insights for clinicians in improving therapeutic outcomes for individuals with heart failure. The findings are limited by the complexity and feasibility of assessing right ventricular dyssynchrony indexes in routine clinical practice.

### 2.4. STE and Mitral Regurgitation after CRT

Matsumoto et al. utilized speckle tracking radial strain to make significant observations about LV dyssynchrony and the correlation with mitral regurgitation (MR) [33]. Their study revealed that patients undergoing CRT and experiencing MR demonstrated enhanced functionality of the mitral apparatus (Table 1). While the precise mechanism remains incompletely elucidated, both global and local remodeling of the LV appears to influence MR. Patients undergoing CRT exhibited a reduction in MR during their long-term follow-up period, coinciding with reverse LV remodeling and, finally, an improvement in LVEF. The study’s retrospective design limited the generalizability and accuracy of its findings.

### 2.5. 3D-STE

Yu Kang and colleagues used 3D-STE to evaluate the LV systolic function and mechanical dyssynchrony in patients with DCM who underwent CRT [34]. The study involved 52 patients with DCM and 55 healthy individuals (Table 1). Echocardiography was performed before and 1 month after the CRT implantation to assess LV dyssynchrony by measuring the standard deviation of time to peak negative value of 3D-STE and the standard deviation of the time to reach the minimum regional volume of 16 segments related to the heart cycle. The study authors reported that 3D-STE displayed high levels of reproducibility and reliability, both within and between observers. Patients with DCM had significantly lower global 3D-STE values compared to healthy individuals.

### 2.6. STE in Optimizing Lead Placement for CRT

STE is widely acknowledged as a significant tool in improving the effectiveness of CRT by aiding in accurate lead placement. The pivotal Targeted Left Ventricular Lead Placement to Guide Cardiac Resynchronization Therapy (TARGET) trial was a significant study in this field [35]. The aim was to determine whether using STE-guided LV lead implantation, which targets areas with the most mechanical dyssynchrony, could result in higher clinical response rates and reverse remodeling effects compared to traditional methods. The study attempted to identify the most effective places for placing the LV lead by using advanced STE metrics such as time to peak strain or strain rate, which are related with the highest level of mechanical dyssynchrony. The findings provide strong evidence in favor of the idea that precise lead placement guided by STE might improve the effectiveness of CRT, leading to the development of more customized and efficient CRT techniques.

Simultaneously, the Speckle Tracking Assisted Resynchronization Therapy for Electrode Region (STARTER) study experiment investigated the involvement of STE in CRT lead placement, with the goal of improving implantation techniques to enhance CRT results [36]. The experiment attempted to improve the accuracy of lead placement in CRT by utilizing insights from STE to understand myocardial mechanical dyssynchrony. The goal was to increase CRT response rates and promote positive reverse remodeling effects. The STARTER experiment attempted to enhance the effectiveness of CRT and improve patient outcomes by carefully incorporating STE data into the process of implanting the CRT lead, thereby refining the art of lead placement.

The combined findings from the TARGET and STARTER trials highlight the significant impact of STE in changing procedures for placing CRT leads. By utilizing STE’s capacity to identify areas with the greatest mechanical dyssynchrony, these trials have advanced the industry towards personalized and focused strategies for CRT, showing a potential for higher clinical response rates and better reverse remodeling outcomes. These trials are important steps in the development of CRT optimization, demonstrating the crucial role of STE in directing the placement of leads and eventually improving patient care in the field of heart failure therapy.

### 2.7. STE in the Assessment of the Left Atrium after CRT

Another study conducted by the research team of D’Andrea aimed to investigate the differential effects of CRT on the left atrial (LA) function in patients with idiopathic DCM versus those with ischemic dilated cardiomyopathy [37]. The study involved 90 patients with heart failure, 47 diagnosed with DCM and 43 with IDCM (Table 1). All patients underwent CRT and were assessed with STE to evaluate the LA function. The patients were evaluated with 2DSE analysis of an atrial longitudinal strain in the basal segments of the LA septum and LA lateral wall and in the LA roof before and 6 months after implantation. The study’s results suggest that the response to CRT regarding the LA function differs between patients with DCM and IDCM. At follow-up, 64 patients (71.1%) (37 idiopathic and 27 ischemic) were responders, and 26 (28.9%) (10 idiopathic; 16 ischemic) were non-responders to CRT. A significant improvement in the LA systolic function was observed only in patients with ischemic DCM responders to CRT. Multivariable analysis found that the ischemic etiology of DCM and positive response to CRT were the only independent determinants of LA lateral wall systolic strain. The study’s findings reveal that CRT had varying effects on the LA function depending on the underlying cause of cardiomyopathy. In patients with IDCM, CRT led to significant LA function improvements, including LA volume reductions and LA strain parameters enhancement. However, in patients with ischemic DCM, CRT did not significantly improve the LA function.

### 2.8. STE in Cardiac Magnetic Resonance Imaging

As the disease progresses, an increasing amount of LV intramyocardial fibrosis is detected. This aspect may interfere with the pacing and sensing thresholds of a CRT. Currently, the best way to evaluate intramyocardial fibrosis is by using cardiac magnetic resonance imaging (cMRI) with late gadolinium enhancement for scar evaluation and T1 mapping for diffuse fibrosis [28]. The longitudinal strain evaluated by the echocardiographic speckle tracking technique has concordant results and is more widely available [28,29,30,31,32,33]. The global longitudinal, circumferential, and radial strain are significantly impaired in patients with DCM. CMR-derived strain analysis is a predictor of adverse events in patients with DCM. Global longitudinal strain analysis has independent and incremental prognostic value for other risk factors, including LVEF, LGE, and ECV [28,29,30,31,32,33]. Peak circumferential strain associated with the absence of LGE and LV mass were found to be predictive of LVRR [28,29,30,31,32,33].

**Table 1 diagnostics-14-01178-t001:** Inclusion and exclusion criteria and clinical characteristics of the participants.

Author/Country/Year	Type of Study/Number of Centers	Number of Subjects/Age	Inclusion Criteria	Exclusion Criteria	Definition of Response
D’Andrea [37]; Italy; 2007	Prospective; single center	90: 47 DCM/43 IDCM; 52.4 ± 10.2 yr	NYHA III–IV; OMT; QRS > 120 ms; LVEF ≤ 35%; LVEDD > 55 mm; SR	Acute HF; severe valve stenosis or valve surgery; MI or CABG < 3 months ago	≥15% ↓ in LV ESV
Carasso [24]; Canada; 2009	Retrospective; single center	76; 65 ± 4 yr	>18 yr; NYHA class III-IV; OMT; QRS > 130 ms; LVEF ≤ 35%	Acute HF; severe valve disease or valve surgery; constrictive pericarditis; uncorrected CHD; ACS or stroke < 6 weeks ago; CABG < 3 months ago; life expectancy < 1 year; pregnancy	≥10% ↑ in LVEF or ≥15% ↓ in LV ESV
D’Andrea [31]; Italy; 2009	Retrospective; single center	110: 60 DCM/50 IDCM; 55.4 ± 11.2 yr	NYHA III–IV; OMT; QRS > 120 ms; LVEF ≤ 35%; LVEDD > 55 mm; SR	Acute HF; right heart HF; ascites; severe valve disease or valve surgery; MI or CABG < 3 months ago; pulmonary embolism; chronic cor pulmonale	≥15% ↓ in LV ESV
Mele [26]; Italy; 2009	Prospective; Multicenter	60 DCM; 68 ± 7 yr	>18 yr; NYHA class III–IV; OMT; QRS > 120 ms; LVEF ≤ 35%; SR	Acute HF; severe valve disease or valve surgery; constrictive pericarditis; uncorrected CHD; ACS or stroke < 6 weeks ago; CABG < 3 months ago; life expectancy < 1 year; pregnancy	20% ↑ in LVEF or ≥15% ↓ in LVESV
Phillips [25]; USA; 2009	Retrospective; single center	100: 47 DCM (57 ± 1 yr)/53 IDCM (69 ± 1 yr)	NYHA III–IV; OMT; QRS > 120 ms; LVEF ≤ 35%	n/a	≥15% ↓ in LVESV
Matsumoto [33]; Japan; 2011	Retrospective; n/a	84: 40 controls (70 ± 9 yr)/44 DCM (70 ± 7 yr)	NYHA III–IV; OMT; QRS > 130 ms; LVEF ≤ 35%; moderate-severe MR	CAD; other causes of cardiomyopathy; morphologic abnormalities of the MV	≥15% ↓ in LVESV
Vitarelli [32]; Italy; 2011	Prospective; single center	81; responders (65 ± 13 yr): 29 IDCM/21 DCM/non-responders: (63 ± 16 yr) 21 IDCM/10 DCM	NYHA III–IV; OMT; LBBB; QRS ≥ 120 ms; LVEF ≤ 35%	myocardial infarction (<3 months); extensive ventricular scars (>4 segments); no OMT; AF	≥15% ↓ in LVESV
Kang [34]; China; 2012	Prospective; single center	107: 55 controls (59.3 ± 12.5 yr)/52 DCM (62.2 ± 13.1 yr)	NYHA III or IV; EF ≤ 35%; QRS ≥ 120 ms, on OMT.	AF; valvular heart disease or CHD; CAD	↑ SDI
Sipula [27]; Czech Republic; 2019	Prospective; single-center	60: 31 IDCM, 29 DCM; 68.5 (31–85) yr	NYHA III/IV, OMT, LVEF < 30%, QRS > 130 ms	n/a	NYHA improvement ≥1 category
Cimino [28]; Italy; 2019	Prospective; single-center	24; 71 ± 11 yr	QRS > 130 ms; OMT; NYHA III or IV; EF ≤ 35%	previous pacemaker implantation; AF & significant valvular disease	>10% ↓ in LVESV
Pescariu [29]; Romania; 2021	Prospective; single-center	44: 17 IDCM, 31 DCM; 64 (53–70) yr	Symptoms despite OMT; LBBB; NYHA II–IV; LVEF ≤ 35%	n/a	CRT parameters: sensing & pacing
Gurgu [30]; Romania; 2023	Prospective; n/a	62; 62 ± 11 yr	NYHA III–IV; OMT; QRS > 130 ms, LBBB, preserved atrioventricular conduction.	ACS or CAD, other causes of cardiomyopathy, AF, severe comorbidities, noncardiac health conditions that limit physical activity.	>5% ↑ in LVEF & 15% ↓ in LVES/LVDV, ↓ MV grade; ↑ NYHA; ↓ hospitalizations/mortality

DCM: dilated cardiomyopathy; IDCM: ischemic dilated cardiomyopathy; ACS: acute coronary syndrome; CAD: coronary artery disease; HF: heart failure; LVEF: left ventricular ejection fraction; LVESV: left ventricular end systolic volume; AF: atrial fibrillation; CRT: cardiac resynchronization therapy; OMT: optimal medical therapy; NYHA: New York heart association; MV: mitral valve; yr: year.

## 3. Conclusions

DCM is a term that describes the final common pathway of various pathogenic processes and gene–environment interactions. DCM can result from extrinsic triggers such as tachyarrhythmias, hypertension, alcohol, chemotherapy, and inflammation. These triggers can promote a reverse remodeling once removed. Therefore, the term “idiopathic” DCM is disappearing, and research on the complex interaction between environmental factors and genetic background is increasing.

Patients with DCM are typically younger, usually between their 30s and 50s and still in their working age [38]. They also have a solid economic and social background. Compared to patients with other causes of heart failure, people with DCM are usually diagnosed approximately 5–10 years earlier and are more likely to be male (70–78%) [38]. They tend to have fewer associated health problems, such as hypertension, diabetes, atrial fibrillation, or stroke, due to their younger age.

In the past 30 years, there has been a significant improvement in the prognosis of patients with DCM. In the 1980s, the average survival rate was approximately 50% in a 5-year follow-up. However, now, at 10 years of follow-up, the survival rate and the rate of being free from heart transplant is beyond 85% [39,40]. This outcome improvement is due to better characterization of etiological factors, medical management for heart failure, and device treatment. It is crucial to identify the cause of DCM accurately by removing all possible triggers of the disease, such as tachyarrhythmias, hypertension, alcohol, chemotherapy, and inflammation, to promote reverse remodeling [39,40]. As the disease progresses, medication options become limited, and heart failure worsens.

Interventional cardiology comes to the aid of these patients’ employing CRT. According to the ESC guidelines, CRT is a recommended treatment for HFrEF, with LVEF under 35%, regardless of the NYHA class, in patients who are still symptomatic despite optimal medical treatment [41]. Patients with nonischemic etiology show a greater improvement in the LV function and a decrease in the NYHA class after CRT [41]. Whether outcomes of patients with HF in DCM differ from those with other HF etiologies is unresolved. Despite relatively clear guidelines, data from the literature report that two-thirds of patients who meet the eligibility criteria have a positive response to the treatment [42,43]. Approximately 30–35% of patients are typically unresponsive to CRT treatment [42,43]. The lack of response to therapy has consistently been a significant weakness of CRT, presenting significant medical and financial difficulties. Identification of the phenotype of an “ideal” CRT responder remains a challenge.

We explored the predictive value STE in determining the response to CRT among patients with DCM. Carasso et al. found that non-responders had notably lower baseline strain values, particularly in lateral, inferior, and anterior regions [24]. Mele et al. showed a finding that baseline strain variability was a better predictor of CRT response than velocity parameters [26]. Sipula et al. reported that STE was the best predictor of improvement in NYHA classification [27]. Cimino et al. showed that improvements in LVEF and GLS were associated with a positive response [28]. Lastly, Pescariu et al. examined the relationship between echocardiographic parameters and CRT functional parameters, and suggested that STE could guide lead positioning and device programming for a better CRT response. Limitations across the studies are the small sample sizes and potential biases introduced by complex echocardiographic techniques.

Continued research on accurately predicting a CRT response suggests the need for the adoption of a holistic approach to the problem of a CRT responder selection, which should include an evaluation of a wide range of cardiovascular disease risk factors. The utilization of artificial intelligence (AI) models may lead to a significant breakthrough in predicting healthcare outcomes. Research has shown that artificial intelligence (AI) has the potential to match or surpass the abilities of healthcare professionals in accurately categorizing illnesses through the analysis of medical imaging [44]. Furthermore, a wide variety of complex data types that are too complicated for human-based analysis may be combined, analyzed, and interpreted by AI [45,46]. Supervised AI models trained on cohorts with over 100 patients obtained up to 85% accuracy and AUC of 0.86 of CRT response prediction for echocardiographic and clinical outcomes, respectively, compared with the guideline-based CRT response prediction accuracy of 70% [46].

## Data Availability

No new data were created or analyzed in this study. Data sharing is not applicable to this article.

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
