# Peer review of "Speckle Tracking Echocardiography in Patients with Non-Ischemic Dilated Cardiomyopathy Who Undergo Cardiac Resynchronization Therapy: A Narrative Review"

_diagnostics, 2024, doi:10.3390/diagnostics14111178_

Round 1

Reviewer 1 Report

Comments and Suggestions for Authors

I have the following comments:

-This is a narrative review of the literature on speckle-tracking in patients with nonischemic DCM undergoing CRT. I strongly suggest that, if possible, it be transformed into a systematic review based on a rigorous literature search based on specific criteria. In this case, a separate paragraph should be added between Sections 1 and 2 reporting on which databases the search was carried out, the time range of the search, the search terms, and the exact inclusion and exclusion criteria for the selection of articles to be finally evaluated. A PRISMA flow chart should also be included illustrating the various steps of the search strategy. A systematic review would add much scientific value and citation potential to the manuscript.

If a systematic review cannot be performed, it should be specified at the end of the Introduction (lines 73-74) that this was a narrative review.

-The section numbering is unclear - I see Section 2 followed by Section 5. Please correct. Moreover, Section 2 basically contains a summary of the main findings of each relevant literature article. It would be important to also highlight any similarities and differences among different studies, and to critically comment on the impact of the various findings on current and potentially future knowledge and practice.  

-At lines 70-72, please expand on the current limitations of imaging (ultrasound-based and not) in the management of patients with DCM undergoing CRT, including cardiac MRI as well.

-There are multiple references to Table 1 throughout the manuscript (e.g., at lines 80, 101, 109 and so on). Please refer to Table 1 only once at the beginning.

-The manuscript should undergo some minor English language editing (e.g., the term 'iscemic' should be replaced by 'ischemic').

Comments on the Quality of English Language

Some minor English language editing should be performed.

Author Response

I would like to thank the reviewer for the comments

This is a narrative review of the literature on speckle-tracking in patients with nonischemic DCM undergoing CRT. I strongly suggest that, if possible, it be transformed into a systematic review based on a rigorous literature search based on specific criteria. In this case, a separate paragraph should be added between Sections 1 and 2 reporting on which databases the search was carried out, the time range of the search, the search terms, and the exact inclusion and exclusion criteria for the selection of articles to be finally evaluated. A PRISMA flow chart should also be included illustrating the various steps of the search strategy. A systematic review would add much scientific value and citation potential to the manuscript.

If a systematic review cannot be performed, it should be specified at the end of the Introduction (lines 73-74) that this was a narrative review.

Unfortunately, a systematic review is not possible to be performed. For this reason, we used the term narrative review.

-The section numbering is unclear - I see Section 2 followed by Section 5. Please correct. Moreover, Section 2 basically contains a summary of the main findings of each relevant literature article. It would be important to also highlight any similarities and differences among different studies, and to critically comment on the impact of the various findings on current and potentially future knowledge and practice.  

The appropriate corrections have been made.

-At lines 70-72, please expand on the current limitations of imaging (ultrasound-based and not) in the management of patients with DCM undergoing CRT, including cardiac MRI as well.

The appropriate corrections have been made.

-There are multiple references to Table 1 throughout the manuscript (e.g., at lines 80, 101, 109 and so on). Please refer to Table 1 only once at the beginning.

The appropriate corrections have been made.

-The manuscript should undergo some minor English language editing (e.g., the term 'iscemic' should be replaced by 'ischemic').

The appropriate corrections have been made.

Reviewer 2 Report

Comments and Suggestions for Authors

Title: SPECKLE TRACKING ECHOCARDIOGRAPHY IN PATIENTS WITH NON-ISCHEMIC DILATED CARDIOMYOPATHY WHO UNDERGO CARDIAC 

RESYNCHRONIZATION THERAPHY

The review should briefly present the echocardiographic techniques for the evaluation of mechanical dyssynchrony (since it was mentioned the PROSPECT study in which TDI was used) and the advantages/disadvantages of speckle tracking echo technique. 

The review should be rewritten, with a better defined structure: 

-       the parameters used for dyssynchrony assessment and the corresponding studies

-        the role of STE in guiding CRT lead placement; The TARGET (Targeted Left Ventricular Lead Placement to GuideCardiacResynchronisationTherapy)trial and The STARTER (Speckle Tracking Assisted Resynchronisation Therapy for Electrode Region) trial are not mentioned

-       the role of STE in right ventricular dyssynchrony 

-       the 3D speckle-tracking echocardiographic data and comparing data obtained from 3D vs 2D techniques

The comments about speckle-tracking strain in cardiac magnetic resonance (CMR) should be included in the text before the conclusions.

            There are many spelling mistakes and unexplained or inconsistent           abbreviations

             Table instead of Table 1 ( it is a single table)

             References must be written in the same way

Comments on the Quality of English Language

The quality of English Language should be improved

Author Response

I would like to thank the reviewer for the comments

The review should briefly present the echocardiographic techniques for the evaluation of mechanical dyssynchrony (since it was mentioned the PROSPECT study in which TDI was used) and the advantages/disadvantages of speckle tracking echo technique. 

 The appropriate corrections have been made.

The review should be rewritten, with a better defined structure: 

-       the parameters used for dyssynchrony assessment and the corresponding studies

The appropriate corrections have been made.

-        the role of STE in guiding CRT lead placement; The TARGET (Targeted Left Ventricular Lead Placement to GuideCardiacResynchronisationTherapy)trial and The STARTER (Speckle Tracking Assisted Resynchronisation Therapy for Electrode Region) trial are not mentioned

-       the role of STE in right ventricular dyssynchrony 

The appropriate corrections have been made.

-       the 3D speckle-tracking echocardiographic data and comparing data obtained from 3D vs 2D techniques

 The appropriate corrections have been made.

The comments about speckle-tracking strain in cardiac magnetic resonance (CMR) should be included in the text before the conclusions.

The appropriate corrections have been made.

            There are many spelling mistakes and unexplained or inconsistent           abbreviations

             Table instead of Table 1 ( it is a single table)

             References must be written in the same way

The appropriate corrections have been made.

Reviewer 3 Report

Comments and Suggestions for Authors

The interesting review by Antoniou et al explored the application of the innovative speckle tracking technique in dilated cardiomyopathy, in order to better select patients who will benefit the most from cardiac resynchronization therapy and understand its prognostic role.

I have some questions for the authors:

1) the definition of DCM is not very clear in the introduction (which is also repeated on page 1, lines 33 and 37), using a different acronym compared to the abstract. To be reformulated, it risks to result confusing for the reader;

2) paragraph 2 is titled "speckle tracking in NIDCM, but then population affected by ischemic heart disease are also covered (for example, the STARTER study, page 4, line 172).

3) in paragraph 2 there is no logical order in the exposition of the contents of the various articles examined. I would try to unify the discussion, or rather introduce the topic covered in each individual paragraph with a title;

4) well-organised table 1, in my opinion a column is missing with the patients excluded based on the inclusion criteria and the feasibility of spekle tracking (I do not think it was always 100%)

Comments on the Quality of English Language

English language is fine, minor editing are required only in logical steps

Author Response

The interesting review by Antoniou et al explored the application of the innovative speckle tracking technique in dilated cardiomyopathy, in order to better select patients who will benefit the most from cardiac resynchronization therapy and understand its prognostic role.

I have some questions for the authors:

  • the definition of DCM is not very clear in the introduction (which is also repeated on page 1, lines 33 and 37), using a different acronym compared to the abstract. To be reformulated, it risks to result confusing for the reader;

The appropriate corrections have been made.

  • paragraph 2 is titled "speckle tracking in NIDCM, but then population affected by ischemic heart disease are also covered (for example, the STARTER study, page 4, line 172).

We included the STARTER study because included pts with DCM and IDCM

3) in paragraph 2 there is no logical order in the exposition of the contents of the various articles examined. I would try to unify the discussion, or rather introduce the topic covered in each individual paragraph with a title;

Extensive changes have been made.

4) well-organised table 1, in my opinion a column is missing with the patients excluded based on the inclusion criteria and the feasibility of spekle tracking (I do not think it was always 100%)

All these information are not available in the majority of the studies, for this reason comment were aided in the main text

Reviewer 4 Report

Comments and Suggestions for Authors

In my opinion, this article is devoted to a current cardiological problem.

However, in my opinion, the specification of the conclusions and introduction could be considered by the authors to improve the perception of the article.

I also recommend refreshing the list of references, since half of the sources are older than the 5-year period.

Author Response

I would like to thank the reviewers for their time and their comments

In my opinion, this article is devoted to a current cardiological problem.

However, in my opinion, the specification of the conclusions and introduction could be considered by the authors to improve the perception of the article. Extensive changes have been made.

I also recommend refreshing the list of references, since half of the sources are older than the 5-year period.                                                                      Extensive changes have been made.

Round 2

Reviewer 1 Report

Comments and Suggestions for Authors

Thank you for your reply.